# Pulmonary Pleomorphic Carcinoma Harboring EGFR Mutation Successfully Treated with Osimertinib: A Case Report

**DOI:** 10.3390/medicina58060706

**Published:** 2022-05-26

**Authors:** Yukari Kano, Nobutaka Kataoka, Yusuke Kunimatsu, Rei Tsutsumi, Izumi Sato, Mai Tanimura, Takayuki Nakano, Keiko Tanimura, Takayuki Takeda

**Affiliations:** Department of Respiratory Medicine, Japanese Red Cross Kyoto Daini Hospital, Kyoto 602-8026, Japan; yukari140030@gmail.com (Y.K.); nkataoka@koto.kpu-m.ac.jp (N.K.); ky92020223@yahoo.co.jp (Y.K.); r-sai@koto.kpu-m.ac.jp (R.T.); izumi-h@koto.kpu-m.ac.jp (I.S.); mai.tanimura@outlook.jp (M.T.); tnakano@koto.kpu-m.ac.jp (T.N.); keiko-t@koto.kpu-m.ac.jp (K.T.)

**Keywords:** brain metastasis, carcinomatous pericarditis, epidermal growth factor receptor, osimertinib, pulmonary pleomorphic carcinoma

## Abstract

Pulmonary pleomorphic carcinoma (PPC) is well-known for its aggressive nature that is usually resistant to platinum-based chemotherapy. On the other hand, the efficacy of an immune checkpoint inhibitor-based regimen in PPC has been elucidated. PPCs harboring epidermal growth factor receptor (EGFR) mutations are extremely rare, and the efficacy of EGFR-tyrosine kinase inhibitors in PPC is limited compared to their efficacy in EGFR-mutated adenocarcinoma. A 43-year-old female patient presenting with a lung mass with multiple brain metastases, carcinomatous pericarditis, and multiple bone metastases was referred to our department. Transbronchial biopsy confirmed the diagnosis of PPC harboring an EGFR mutation with exon 19 deletion. Subsequently, she was treated with osimertinib, a third-generation EGFR-tyrosine kinase inhibitor, which resulted in partial response with shrinkage of the primary lesion and brain metastases. This partial response remained durable for 11 months with an ongoing regimen. The current case suggests that osimertinib would show promising effects as a first-line treatment for PPCs harboring EGFR mutations, as well as a reasonable sequence of therapy followed by immune checkpoint inhibitor-based regimens.

## 1. Introduction

Pulmonary pleomorphic carcinoma (PPC) is a very rare type of poorly differentiated non-small cell lung cancer (NSCLC) with an aggressive nature [1] that is usually resistant to platinum-based chemotherapy [1,2]. On the other hand, the efficacy of immune checkpoint inhibitor (ICI) monotherapy [3,4,5,6] or chemoimmunotherapy [7] in PPC has been elucidated, which led to improved outcomes. Although epidermal growth factor receptor (EGFR) mutations are found in approximately 15–20% of PPCs [8,9], the efficacy of EGFR-tyrosine kinase inhibitors (EGFR-TKIs) in PPC is limited compared to their efficacy in EGFR-mutated adenocarcinoma [8]. Thus, the optimal sequence of therapy for PPC harboring EGFR mutation remains unclear.

This study reports a case of a patient with PPC harboring an EGFR mutation that was successfully treated with osimertinib, resulting in partial response (PR) with long progression-free survival.

## 2. Case Report

A 43-year-old female patient was referred to our department complaining of dry cough and dyspnea on exertion. She had never smoked and had no past medical history of malignancy. Chest computed tomography showed a mass in the right lower lobe (Figure 1A) with bilateral pleural effusions and pericardial effusion (Figure 1B). Since cardiac function was affected due to the pericardial effusion causing cardiac tamponade, pericardial drainage was immediately performed. Cytological evaluation of the drained fluid demonstrated malignant cells. Brain magnetic resonance imaging with contrast enhancement revealed multiple brain metastases in bilateral cerebral hemispheres and the right midbrain, with the largest one in the left temporal lobe (Figure 1C). Transbronchial biopsy was performed, and hematoxylin-eosin staining of the specimen showed sarcomatoid carcinoma features with giant cell invasion and loose inter-cellular connections (Figure 1D). In addition, it showed adenocarcinoma features with columnar and cylindrical structures (Figure 1E). The specimen collected from the tumor contained more than 10% giant cells. Therefore, the final diagnosis was PPC with multiple brain metastases, bone metastases, and carcinomatous pericarditis (cT4N3M1c [BRA, OSS, OTH], stage IVB). Thereafter, EGFR mutation with exon 19 deletion was detected, and the programmed cell death-ligand 1 (PD-L1) expression rate on tumor cells was found to be <1%.

Osimertinib, a third-generation EGFR-TKI, was administered at a dose of 80 mg daily as the first-line treatment. Subsequently, shrinkage of the primary lesion was observed 25 days after treatment (Figure 2A). Although the right pleural effusion increased in amount at the first evaluation (Figure 2A), it decreased three months after osimertinib treatment (Figure 2B) and almost disappeared after seven months (Figure 2C). Multiple brain metastases also showed PR after 24 days (Figure 2D), while further shrinkage was observed after three months (Figure 2E). The primary lesion, multiple brain metastases, and carcinomatous pericarditis showed PR for 11 months after osimertinib treatment, which is ongoing without any symptoms related to the PPC.

## 3. Discussion

PPC is a very rare type of NSCLC that accounts for 0.1–1.6% of all thoracic malignancies [1,10]. PPC is classified as a sarcomatoid carcinoma, characterized by a composite of sarcomatoid (spindle and/or giant cell elements) and epithelial (adenocarcinoma, squamous cell carcinoma, or undifferentiated NSCLC) components, with spindle and/or giant cells comprising at least 10% of the tumor cells [11]. This complex histological feature is considered a consequence of the epithelial-mesenchymal transition of the epithelial component [12]. Contextually, the fact that adenocarcinoma with epithelial-mesenchymal transition possesses high PD-L1 expression [13] could explain the previously reported high PD-L1 expression in PPC ranging from 69.2 to 90.2% [14,15].

In addition to the lack of an established treatment strategy for PPC due to its rareness, the available scientific evidence remains even more scarce for the treatment of PPC harboring EGFR mutation.

Although the efficacy of platinum-based chemotherapy for PPC is unsatisfactory leading to poor prognosis [1,2], the efficacy of ICI monotherapy [3,4,5,6] or chemoimmunotherapy [7] has been elucidated. The efficacy of these treatment regimens is linked to the abovementioned high expression of PD-L1 in PPC.

On the other hand, the efficacy of EGFR-TKIs for PPC harboring EGFR mutation is reportedly limited, and it depends on the extent of EGFR oncogene dependence on the malignant cells [8]. The concept of intratumor heterogeneity in tumors [16,17] could explain this mechanism. Furthermore, the relationship between the heterogeneity of EGFR mutation among lung adenocarcinoma and the mixed response to EGFR-TKIs would be a good explanation for this observation [18]. Since the genetic testing results of EGFR mutations are not quantitatively but qualitatively reported, it is difficult to predict the extent of oncogene dependence for PPC with EGFR mutation. In other words, the efficacy of EGFR-TKI is unpredictable.

Thus, the decision-making process regarding the treatment sequence for PPC harboring EGFR mutation is difficult. In other words, it is challenging to determine which treatment option, between EGFR-TKIs and ICI-based regimens, should be the first-line treatment, owing to the lack of concrete evidence regarding the superiority of the two regimens. If ICI-based treatment is selected as the first-line therapy, durable immune-related adverse events could be an obstacle in the subsequent second-line treatment with EGFR-TKIs, with concerns about drug-induced interstitial lung disease (DI-ILD). Indeed, the concomitant administration of osimertinib and durvalumab (anti-PD-L1 antibody) for treatment-naïve NSCLC harboring EGFR mutation resulted in DI-ILD in 20.0% of the patients that received durvalumab 3 mg/kg once every 2 weeks and in 23.1% of the patients that received a dose of 10 mg/kg once every 2 weeks in the TATTON phase Ib trial [19]. On the other hand, this combination treatment for EGFR-T790M-positive NSCLC patients resulted in DI-ILD in only 3% of the participants in the CAURAL phase III trial [20]. While the reason behind the discrepancy in the observed incidence of DI-ILD remains unclear, concomitant administration of osimertinib and durvalumab should be avoided. Because there is no evidence on the period of time for which EGFR-TKIs would be safely administered after ICI treatment, it is preferred that EGFR-TKIs be administered prior to ICI-based treatment. This explains why osimertinib was selected as the first-line treatment in the current case, which resulted in PR and long progression-free survival.

Although the management of oligoprogression such as brain metastases and bone metastases during osimertinib treatment in PPC harboring EGFR mutation has not been established, it should be in accordance with the established treatment strategy in EGFR-mutant lung adenocarcinoma; local consolidative therapy mainly with radiotherapy, without cessation of osimertinib [21].

In the case of systemic progression after osimertinib, the second-line treatment would be chemoimmunotherapy in the current case. ICI monotherapy would be out of the question because hyperprogressive disease following ICI monotherapy for PPC harboring EGFR mutation has been reported [22]. The preferred chemoimmunotherapy regimen would be atezolizumab in combination with bevacizumab, carboplatin, and solvent-based paclitaxel, which has demonstrated significant efficacy on patients with NSCLC harboring EGFR mutation [23].

## 4. Conclusions

The observed outcomes in the current case suggest that osimertinib would show promising effects as a first-line treatment for PPCs harboring EGFR mutations, while its efficacy depends on the extent of EGFR oncogene dependence of the tumor. Although ICI-based treatment is promising for PPCs, the risk of DI-ILD should be taken into account when ICI-based treatment is administered prior to EGFR-TKIs. Thus, osimertinib as a first-line treatment would provide a reasonable sequence of therapy followed by ICI-based regimens.

## Figures and Tables

**Figure 1 medicina-58-00706-f001:**
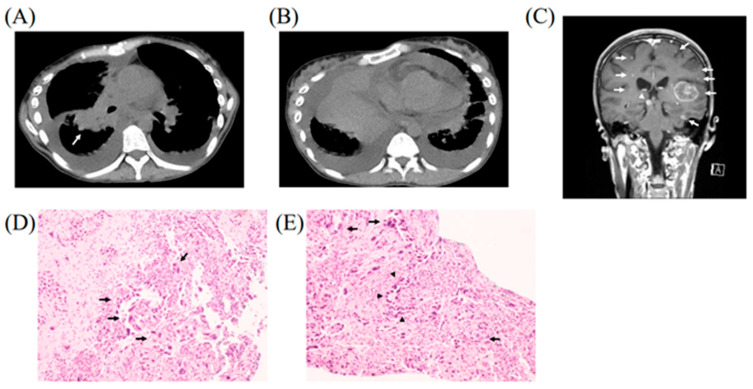
Chest computed tomography showed a mass in the right lower lobe (arrow) accompanied by hilar and mediastinal lymph nodes metastases (**A**), bilateral pleural effusions, and a pericardial effusion (**B**). Brain magnetic resonance imaging with contrast enhancement (coronal T1-weighted image) revealed multiple brain metastases in bilateral cerebral hemispheres (arrows) and right midbrain (arrowhead), with the largest metastasis in the left temporal lobe (**C**). Hematoxylin-eosin staining of the transbronchial biopsy specimen showed giant cell invasion (arrows) suggesting sarcomatoid carcinoma (**D**,**E**) as well as columnar and cylindrical structures (arrowheads) suggesting adenocarcinoma component (**E**). More than 10% of the cells in the specimen were giant cells (**D**).

**Figure 2 medicina-58-00706-f002:**
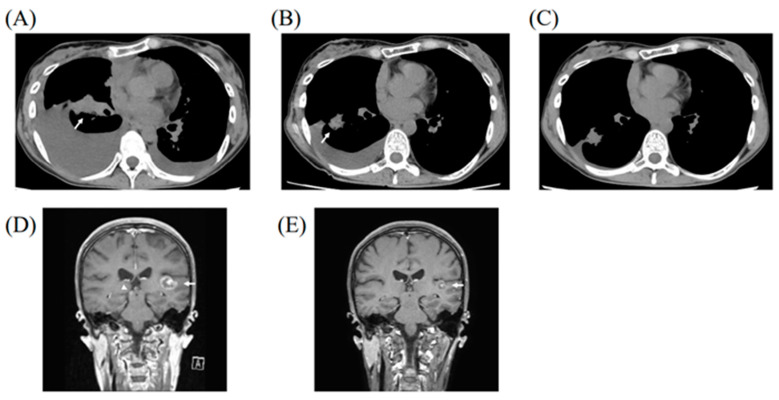
Chest computed tomography taken 25 days after initiation of osimertinib treatment showed shrinkage of the primary lesion (arrow), with increased right pleural effusion (**A**). The primary lesion (arrow) continued shrinking after three months of treatment, with right pleural effusion decreased in amount (**B**), which almost disappeared seven months after treatment (**C**). Multiple brain metastases also showed significant shrinkage in the left temporal lobe (arrow) and right midbrain (arrowhead), with a disappearance in the remaining lesions after 24 days (**D**). Further shrinkage of the brain metastases was observed after three months (**E**).

## Data Availability

The data supporting the conclusion are included in the article.

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
