# Peer review of "Pulmonary Pleomorphic Carcinoma Harboring EGFR Mutation Successfully Treated with Osimertinib: A Case Report"

_medicina, 2022, doi:10.3390/medicina58060706_

Round 1
Reviewer 1 Report
- The English need improvement since there are some grammatical and syntax errors in the manuscript. For example,
- in line number 12, the words “an aggressive” may be as “its aggressive”;
- in line number 13, “immune” as “an immune”;
- in line number 42, “a pericardial” as “pericardial”;
- in line number 43, “a cardiac” as “cardiac”;
- in line number 82, “regimens is” as “regimens are”.
The grammar mistakes which are not mentioned here are also to be checked and corrected properly.
- Check the abbreviations throughout the manuscript and introduce the abbreviation when the full word appears the first time in the text and then use only the abbreviation (For example, partial response, etc.). And it should be in both abstract as well as in the remaining part of the manuscript. Make a word abbreviated in the article that is repeated at least three times in the text, not all words need to be abbreviated.
- The intrudouction part appears less informative about the pulmonary pleomorphic carcinoma, thus this section should be indicated as detailed to understand the manuscript in clear.
- The conclusion section appears to be just a detailed summary of results/observations. All conclusions must be convincing statements on what was found to be novel, and impactful based on the strong support of the data/results/discussion. The authors may add a section to the manuscript regarding the future work.
Author Response
I really appreciate your sincere advice and suggestions for our manuscript.
I have read through your comments and critiques, and revised our manuscript in accordance with your comments.
I believe that the manuscript has become sophisticated after this revision, and has become suitable for the publication in Medicina.
Comments and Suggestions for Authors
- The English need improvement since there are some grammatical and syntax errors in the manuscript. For example,
- in line number 12, the words “an aggressive” may be as “its aggressive”;
- in line number 13, “immune” as “an immune”;
- in line number 42, “a pericardial” as “pericardial”;
- in line number 43, “a cardiac” as “cardiac”;
- in line number 82, “regimens is” as “regimens are”.
The grammar mistakes which are not mentioned here are also to be checked and corrected properly.
Response:
I really appreciate your suggestions, and I have carefully read through the manuscript with a native speaker.
As to the part which you mentioned above (in line number 82, “regimens is” as “regimens are), I think the subject of this sentence is “The efficacy”. Thus, I think the grammar is correct in this part.
- Check the abbreviations throughout the manuscript and introduce the abbreviation when the full word appears the first time in the text and then use only the abbreviation (For example, partial response, etc.). And it should be in both abstract as well as in the remaining part of the manuscript. Make a word abbreviated in the article that is repeated at least three times in the text, not all words need to be abbreviated.
Response:
I really appreciate your suggestions, and I have carefully read through the manuscript.
I deleted those abbreviations which are repeated only twice: “ICI” and “TKI” in the Abstract, “PFS” in the Introduction and the Discussion, “H-E” in the Case Report, “EMT” in the Discussion section.
- The intrudouction part appears less informative about the pulmonary pleomorphic carcinoma, thus this section should be indicated as detailed to understand the manuscript in clear.
Response:
I really appreciate your sincere advice, and added two sentences in the Introduction so as to enrich the contents; “On the other hand, the efficacy of immune checkpoint inhibitor (ICI) monotherapy (3-6) or chemoimmunotherapy (7) in PPC has been elucidated, which led to the improved outcomes.” and “Thus, the optimal sequence of therapy for PPC harboring EGFR mutation remains unclear”.
- The conclusion section appears to be just a detailed summary of results/observations. All conclusions must be convincing statements on what was found to be novel, and impactful based on the strong support of the data/results/discussion. The authors may add a section to the manuscript regarding the future work.
Response:
I really appreciate your suggestion, and added some expressions to clarify the importance of a sequence of therapy of first-line osimertinib treatment followed by ICI-based regimens, which we wanted to express in this article.

Reviewer 2 Report
Nicely written care report. Minor comments are suggested. Attached.

Author Response
Comments and Suggestions for Authors
Nicely written care report. Minor comments are suggested. Attached.
What happened after that? till the current time?
Response:
I really appreciate your kind advice for our manuscript.
The efficacy of osimertinib is ongoing and the patient is still enjoying the PFS.
Therefore, I added a description “, which is ongoing” to the sentence which you pointed out. And I also added a description “with an ongoing regimen” in the Abstract.
